# BAFF-neutralizing interaction of belimumab related to its therapeutic efficacy for treating systemic lupus erythematosus

Woori Shin[1], Hyun Tae Lee[1], Heejin Lim[1], Sang Hyung Lee[1], Ji Young Son[1], Jee Un Lee[1], Ki-Young Yoo[1], Seong Eon Ryu[2], Jaejun Rhie[1], Ju Yeon Lee[1] & Yong-Seok Heo[1]

BAFF, a member of the TNF superfamily, has been recognized as a good target for auto-immune diseases. Belimumab, an anti-BAFF monoclonal antibody, was approved by the FDA for use in treating systemic lupus erythematosus. However, the molecular basis of BAFF neutralization by belimumab remains unclear. Here our crystal structure of the BAFF–belimumab Fab complex shows the precise epitope and the BAFF-neutralizing mechanism of belimumab, and demonstrates that the therapeutic activity of belimumab involves not only antagonizing the BAFF–receptor interaction, but also disrupting the for-mation of the more active BAFF 60-mer to favor the induction of the less active BAFF trimer through interaction with the flap region of BAFF. In addition, the belimumab HCDR3 loop mimics the DxL(V/L) motif of BAFF receptors, thereby binding to BAFF in a similar manner as endogenous BAFF receptors. Our data thus provides insights for the design of new drugs targeting BAFF for the treatment of autoimmune diseases.

[1] Department of Chemistry, Konkuk University, 120 Neungdong-ro, Gwangjin-gu, Seoul 05029, Republic of Korea. [2] Department of Bio Engineering, Hanyang University, 222 Wangsimni-ro, Seongdong-gu, Seoul 04763, Republic of Korea. These authors contributed equally: Woori Shin, Hyun Tae Lee, Heejin Lim, Sang Hyung Lee. Correspondence and requests for materials should be addressed to Y.-S.H. (email: ysheo@konkuk.ac.kr)

The binding of the tumor necrosis factor superfamily (TNFSF) members to their cognate tumor necrosis factor receptor superfamily (TNFRSF) members initiates many immune and inflammatory processes. Many monoclonal antibodies blocking TNFSF/TNFRSF interactions have been investigated for therapeutic application[1]. The drugs infliximab (Remicade) and adalimumab (Humira), which are monoclonal antibodies targeting TNFα, have enabled the development of breakthrough therapies for the treatment of several autoimmune inflammatory diseases, including rheumatoid arthritis, Crohn's disease, and psoriatic arthritis[2].

B-cell activating factor (BAFF; also known as BLyS, zTNF4, TNFSF13B, THANK, and TALL-1) is a member of TNFSF and a crucial factor for survival and maturation of B cells[3,4]. BAFF is expressed as a homotrimeric transmembrane protein that can be released as a soluble homotrimeric cytokine after cleavage at a furin protease site. Interestingly, soluble BAFF trimers can oligomerize to a virus-like assembly consisting of 20 trimers through trimer–trimer interactions via a long DE loop called a "flap" region, which is unique among TNFSF members[5–7]. This BAFF 60-mer is considerably more active than the trimer, possibly due to the clustering of BAFF receptors on the B-cell surface[8,9]. BAFF can signal through three different receptors on B cells: BAFF receptor 3 (BR3; also known as BAFF-R), B-cell maturation antigen (BCMA), and transmembrane activator and calcium-modulator and cytophilin ligand interactor (TACI)[10–12]. Among these receptors, BR3 is the principal receptor for B-cell survival signaling by BAFF.

Systemic lupus erythematosus (SLE), also known simply as lupus, is a relapsing and idiopathic autoimmune inflammatory disease. Although the cause of this disease is unclear, it has long been accepted that autoantibodies produced by autoreactive B cells have a central function in SLE pathogenesis[13,14]. Mice overexpressing BAFF induce B-cell hyperplasia and display symptoms of autoimmune disorders, whereas BAFF-knockout mice lack mature B cells[15–17]. In addition, autoreactive B cells have a greater dependency on BAFF for their survival as compared with protective B cells[18,19]. Elevated BAFF levels are detected in human patients with SLE; therefore, targeting the BAFF/receptor axis has emerged as a logical therapeutic candidate for control of SLE through the modulation of aberrant autoantibody production by autoreactive B cells[20–22].

In 2011, the FDA approved belimumab (Benlysta), a fully human monoclonal $IgG_1\lambda$ antibody neutralizing soluble BAFF, as the first targeted therapy for SLE in the past 50 years since the introduction of corticosteroids and immunosuppressive therapy[23]. Belimumab has provided clinical benefits to SLE patients by reducing the number of circulating naive B cells, activated B cells, and plasma cells, but not memory B cells or T cells[24]. Furthermore, belimumab treatment does not affect antibody responses to previous pneumococcal, tetanus, or influenza immunizations, which is consistent with preservation of the memory B-cell compartment[25]. In addition to belimumab, other BAFF antagonists, including tabalumab (anti-BAFF antibody), blisibimod (anti-BAFF peptibody), and atacicept (TACI-IgG Fc fusion), are being or have been investigated in clinical trials for SLE[14]. These three biologics differ from belimumab in that they bind both membrane-bound and soluble BAFF, whereas belimumab binds only soluble BAFF[14,26,27].

Crystal structures of BAFF alone or in complex with its cognate receptors have established the structural foundation for the oligomeric state of BAFF and its interaction with receptors[5–7,28]. However, no structure of BAFF in complex with any therapeutic antibody is available, even though the precise mechanism and epitope are crucial elements of antibody drugs. Here we report the 2.05 Å resolution structure of the BAFF–belimumab Fab complex

## Table 1 Data collection and refinement statistics

|  | BAFF/belimumab Fab | Belimumab Fab |
|---|---|---|
| Data collection |  |  |
| X-ray source | PLS 5C | PLS 7A |
| Wavelength (Å) | 1.0000 | 1.0000 |
| Space group | $P2_13$ | $P2_1$ |
| Cell dimensions |  |  |
| $a, b, c$ (Å) | 134.00, 134.00, 134.00 | 60.91, 52.30, 63.75 |
| $\alpha, \beta, \gamma$ (°) | 90, 90, 90 | 90, 90.72, 90 |
| Resolution (Å) | 2.05 (2.09–2.05)[a] | 1.90 (1.93–1.90) |
| $R_{sym}$ (%) | 8.3 (42.4) | 5.5 (25.3) |
| $I/\sigma I$ | 51.6 (3.1) | 41.1 (4.3) |
| Completeness (%) | 96.7 (95.9) | 97.7 (87.8) |
| Redundancy | 14.5 (10.5) | 3.4 (2.6) |
| Refinement |  |  |
| Resolution (Å) | 2.05 | 1.90 |
| No. reflections | 48,698 | 31,056 |
| $R_{work}/R_{free}$ (%) | 18.0/21.8 | 16.7/20.6 |
| No. atoms |  |  |
| Protein | 4331 | 3172 |
| Water | 500 | 393 |
| $B$-factors (Å$^2$) | 38.4 | 29.9 |
| R.m.s. deviation |  |  |
| Bond lengths (Å) | 0.007 | 0.007 |
| Bond angles (°) | 1.129 | 1.096 |
| Ramachandran |  |  |
| Favored (%) | 97.70 | 97.61 |
| Allowed (%) | 2.13 | 2.39 |
| Outlier (%) | 0.18 | 0.00 |
| PDB ID code | 5Y9J | 5Y9K |

[a] Values in parentheses are for the outer resolution shell

and 1.90 Å resolution structure of belimumab Fab alone for elucidating the molecular mechanism of this antibody drug against BAFF. The nature of the antigenic epitope of belimumab is further analyzed by site-directed mutagenesis, and the distinct BAFF-binding mechanisms between belimumab and tabalumab are also investigated by analytical gel-filtration chromatography.

## Results

**Crystal structure of the BAFF–belimumab Fab complex.** In this study, a soluble form of human BAFF (aa 134–285) was expressed in *Escherichia coli*, and the Fab fragments of belimumab and tabalumab were produced by periplasmic expression in *E. coli*. Gel-filtration analysis indicated that BAFF and belimumab Fab form a 3:3 complex in solution. The crystal structure of the BAFF–belimumab Fab complex was determined and refined to 2.05 Å resolution with $R/R_{free} = 0.180/0.218$ (Table 1). The electron density map was clearly shown throughout the entire protein complex, including all the complementarity-determining regions (CDR) of the belimumab Fab fragment (Fig. 1). One BAFF–belimumab complex exists in an asymmetric unit, and the central BAFF trimer is bound by three belimumab Fab molecules through a crystallographic 3-fold axis, which is analogous to the structures of the BAFF–BR3 and BAFF–BCMA complexes[6,7] (Fig. 1). When viewed along the 3-fold axis, the trimeric complex has a shape that resembles a three-bladed propeller, with one protomer representing one blade. The pseudo 2-fold axes of the Fab fragments of bound belimumab intersect the 3-fold axis of the trimeric complex of BAFF–belimumab and form an approximate angle of 50° downward from a plane perpendicular to the 3-fold axis. When we consider a cell with a BAFF precursor attached, this plane is supposed to represent the cell membrane. In this binding orientation, belimumab not only binds soluble

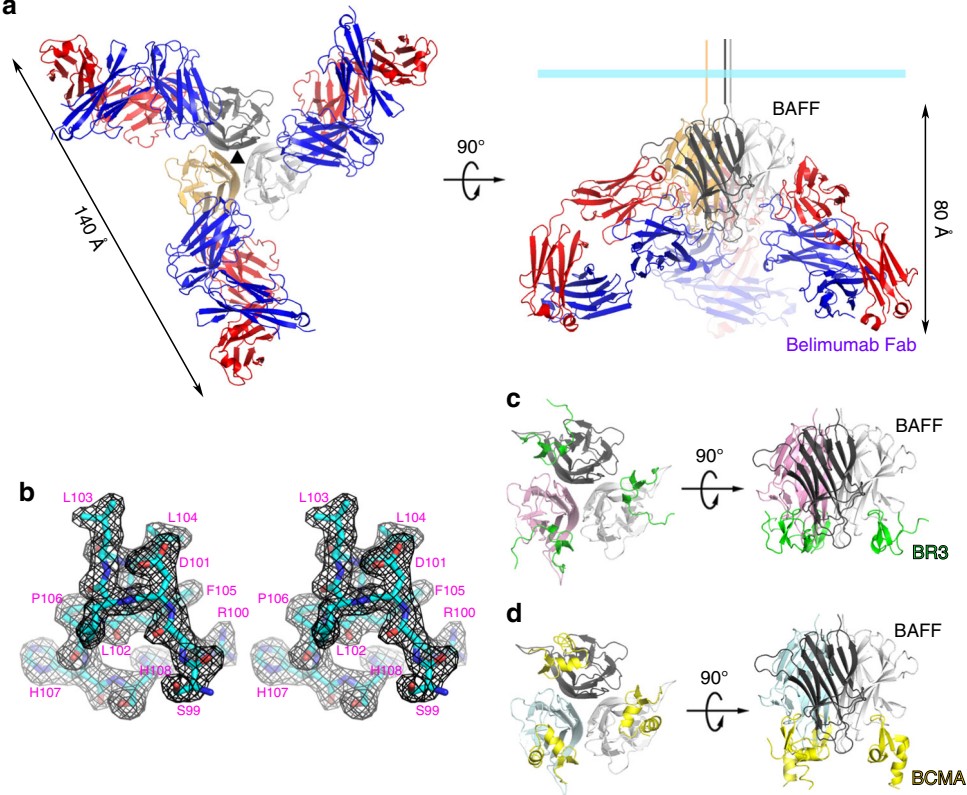

**Fig. 1** Overall structure of BAFF in complex with belimumab Fab. **a** Ribbon representation of the BAFF trimer (yellow, black, gray) in complex with the belimumab Fab fragment (heavy chain: blue; light chain: red) in two orientations. The 3-fold axis of the trimeric complex is indicated as a black triangle. The cyan bar indicates the putative membrane of a BAFF-producing cell if the BAFF trimer is in its membrane-bound form. **b** Stereoview of the $2f_o$–$f_c$ composite omit map (1.2 $\sigma$ contour level) at HCDR3 of belimumab in the complex structure, calculated at 2.05 Å resolution. **c** Structure of the BAFF trimer (pink, black, gray) in complex with three BR3 (green) proteins (PDB ID code 1OQE). **d** Structure of BAFF trimer (cyan, black, gray) in complex with three BCMA (yellow) proteins (PDB ID code 1OQD)

BAFF but also, in principle, might be able to bind membrane-bound BAFF. However, belimumab has been characterized as targeting soluble BAFF only[26]. The reason belimumab cannot bind membrane-bound BAFF remains unclear based on this BAFF–belimumab complex structure. Superimposing BAFF in the BAFF–belimumab complex onto its receptor-bound form yielded a root mean square deviation of 0.27 Å for all Cα atoms and indicated that no significant overall structural change occurred upon complex formation, except for the conformational change in the DE loop, also known as the "flap" region (Fig. 2d and Supplementary Fig. 1). This flap region contributes to the BAFF trimer–trimer interaction, thereby mediating the construction of a virus-like assembly comprising 60 BAFF monomers[5]. Previous structural studies showed that the receptor-binding sites remain exposed and accessible on the surface of the BAFF 60-mer[6,7] (Fig. 2c). As shown in previous studies, analytical gel-filtration chromatography indicated that soluble BAFF exists in equilibrium between its trimeric form and the 60-mer (Fig. 2a). However, belimumab does not bind to the intact BAFF 60-mer, but disassembles it into trimers of BAFF–belimumab complexes, given that the belimumab epitope is partially hidden within the structure of the BAFF 60-mer (Fig. 2a). The LCDR1 and LCDR2 of belimumab interact with the flap region, thereby locking the loop into a conformation that disallows BAFF trimer–trimer interactions (Fig. 2d). By contrast, tabalumab Fab binds to both the 60-mer and trimer without disrupting the oligomeric states of BAFF, implying that its epitope would be completely exposed on the surface of the 60-mer, similar to the receptor-binding sites of BAFF (Fig. 2b).

The crystal structure of the belimumab Fab fragment alone was also determined and refined to 1.90 Å resolution with $R/R_{\text{free}} =$ 0.167/0.206 (Table 1). The elbow angle of the belimumab Fab, defined as the angle subtended by the two pseudo-dyad axes relating the variable and constant domains of a Fab fragment, differed by about 5° between the BAFF-bound and free belimumab structure due to intrinsic elbow flexibility (Supplementary Fig. 2a). Despite the absence of the binding partner (BAFF), the electron density of belimumab was clear throughout the entire structure of the Fab fragment, including all six CDRs. Structural comparison of the belimumab CDR loops before and after binding to BAFF showed little conformational deviation and minor adjustments in the side chains involved in the interaction with BAFF, implying that belimumab maintains the CDR loops in productive conformations prior to binding to BAFF, thereby contributing to the high-affinity binding (Supplementary Fig. 2b).

**Interactions between BAFF and belimumab Fab.** The BR3 or BCMA-binding region of BAFF is unusually limited as compared with other TNFSF-TNFRSF interfaces, with a buried surface area of about 1300 Å² (Fig. 3b, c)[6,7]. However, belimumab Fab binds BAFF through a large and highly complementary interface, with a buried surface area of 2013 Å² (Fig. 3a). The belimumab epitope comprises mainly residues from a single protomer of the BAFF trimer, with an adjacent protomer playing an accessory role in belimumab binding. The belimumab epitope can be divided into two distinct regions (Fig. 3a). Region I nearly encompasses the receptor-binding site on BAFF and consists of several discontinuous segments, including residues $_{\text{BAFF}}$S162, $_{\text{BAFF}}$Y163,

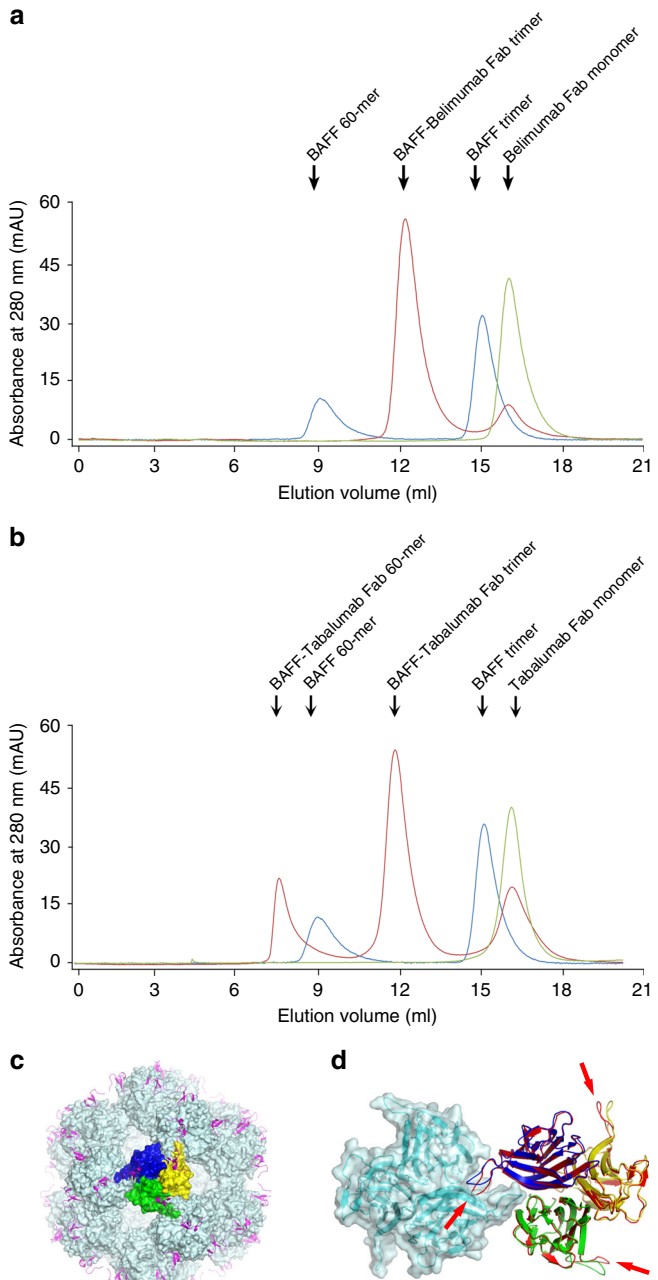

**Fig. 2** Disruption of the BAFF 60-mer by belimumab binding. **a** Analytical gel-filtration chromatography of BAFF alone (blue), belimumab Fab alone (green), and BAFF incubated with excess belimumab Fab (red). **b** Analytical gel-filtration chromatography of BAFF alone (blue), tabalumab Fab alone (green), and BAFF incubated with excess tabalumab Fab (red). **c** Structure of the BAFF 60-mer (cyan) in complex with 60 BR3 (purple) proteins (PDB ID code 4V46). One trimer in the BAFF 60-mer is colored blue, green, and yellow for each respective protomer. **d** Superposition of the BAFF trimers extracted from the structures of the BAFF–belimumab complex (red) and the BAFF 60-mer (blue, green, and yellow). An adjacent trimer in the BAFF 60-mer (cyan) is represented as a mixed ribbon/surface model. The arrows indicate the conformational change in the DE loop upon belimumab binding

$_{BAFF}$Y206, $_{BAFF}$R231, $_{BAFF}$I233, $_{BAFF}$R265, and $_{BAFF}$E266 in a protomer of the BAFF trimer and $_{BAFF}$L240 and $_{BAFF}$N242 in an adjacent protomer (Fig. 4a and Supplementary Fig. 1b). The three CDRs from the heavy chain (HCDR1, HCDR2, and HCDR3) and LCDR3 from the light chain of belimumab participate in the

interaction with region I of BAFF. The belimumab residues involved in the interaction with region I are $_{heavy}$N31 of HCDR1; $_{heavy}$M54, $_{heavy}$F55, and $_{heavy}$T57 of HCDR2; $_{heavy}$D101, $_{heavy}$L102, $_{heavy}$L103, and $_{heavy}$L104 of HCDR3; and $_{light}$N95 of LCDR3 (Fig. 4a and Supplementary Fig. 2c). Several hydrogen bonds are formed by the side chain atoms of $_{BAFF}$Y206, $_{BAFF}$R231, and $_{BAFF}$E266 and the main chain atoms of $_{BAFF}$S162, and a salt bridge is formed by $_{BAFF}$R265 (Fig. 4a). In addition, $_{BAFF}$S162, $_{BAFF}$Y163, $_{BAFF}$Y206, $_{BAFF}$I233, $_{BAFF}$L240, and $_{BAFF}$N242 contribute to van der Waals interactions with belimumab. Region II comprises residues within the BAFF flap region, including $_{BAFF}$D222, $_{BAFF}$L224, and $_{BAFF}$S225 (Fig. 4a and Supplementary Fig. 1b). Belimumab residues involved in the interaction with region II are from the light chain, including $_{light}$R28 and $_{light}$Y31 of LCDR1 and $_{light}$K50 of LCDR2 (Fig. 4a and Supplementary Fig. 2c). $_{BAFF}$D222 forms a salt bridge with $_{light}$K50, and the main chain atoms of $_{BAFF}$S225 and $_{light}$R28 form a hydrogen bond (Fig. 4a). In addition, $_{BAFF}$L224 interacts with $_{light}$Y31 via van der Waals contact.

According to structural analysis of the BAFF–belimumab interactions, we selected 10 residues from BAFF, including, $_{BAFF}$Y163, $_{BAFF}$Y206, $_{BAFF}$D222, $_{BAFF}$L224, $_{BAFF}$R231, $_{BAFF}$I233, $_{BAFF}$L240, $_{BAFF}$N242, $_{BAFF}$R265, and $_{BAFF}$E266, for mutagenesis studies. Each residue was replaced by alanine, and the binding affinities of each mutant with belimumab were measured through surface plasmon resonance (SPR) to evaluate the effects of these residues on the BAFF–belimumab interaction (Fig. 4b and Supplementary Table 1). Substitution of $_{BAFF}$L240 and $_{BAFF}$N242 with alanine hardly affected the binding capacity of BAFF with belimumab, implying that belimumab binding is mediated mainly by residues from a single protomer of the BAFF trimer, and that the contribution to binding affinity by an adjacent BAFF protomer is marginal. The hydrogen bonds formed by $_{BAFF}$R231 and $_{BAFF}$E266 and hydrophobic interactions involving $_{BAFF}$Y163 and $_{BAFF}$I233 are also not critical to the antigen–antibody interaction. Notably, the replacement of $_{BAFF}$Y206 resulted in a drastic decrease in binding affinity based on a 15-fold higher dissociation constant ($K_D$). The hydrogen bonds and hydrophobic interaction mediated by $_{BAFF}$Y206 are critical to maintaining the stable BAFF–belimumab complex, as the $_{BAFF}$Y206A mutation increased the off-rate constant ($k_{off}$) by 80 fold. The $_{BAFF}$R265A mutation decreased the on-rate constant ($k_{on}$) by 7 fold, implying that the salt bridge formed between $_{BAFF}$R265 and $_{heavy}$D101 contributes much to the rapid association of BAFF and belimumab. These data indicate that the key interactions mediated by region I are concentrated in a very narrow subregion containing two residues, $_{BAFF}$Y206 and $_{BAFF}$R265. The interactions of region II with the light chain of belimumab, including the hydrophobic interaction between $_{BAFF}$L224 and $_{light}$Y31 and the salt bridge between $_{BAFF}$D222 and $_{light}$K50, are also important for slow dissociation, given that the $_{BAFF}$L224A and $_{BAFF}$D222A mutations increased $k_{off}$ by 12 and 4 fold, respectively. Previously, a crystal structure of the BAFF 60-mer showed that $_{BAFF}$L224 contributes to the binding energy associated with BAFF trimer–trimer interactions[5]. Therefore, belimumab binding to region II sterically blocks BAFF trimer–trimer interactions and alters the DE loop conformation into one unfavorable for the formation of a higher-order oligomeric state of BAFF (Figs. 2d and 5). These findings suggest that the neutralizing effect of belimumab is a consequence of not only the direct inhibition of BAFF binding to receptors through interactions within region I, but also the disruption of the 60-mer, which is the most active form of BAFF, through interactions within region II.

**The belimumab HCDR3 loop mimics the DxL(V/L) motif.** The DxL(V/L) motif is completely conserved among BAFF receptors

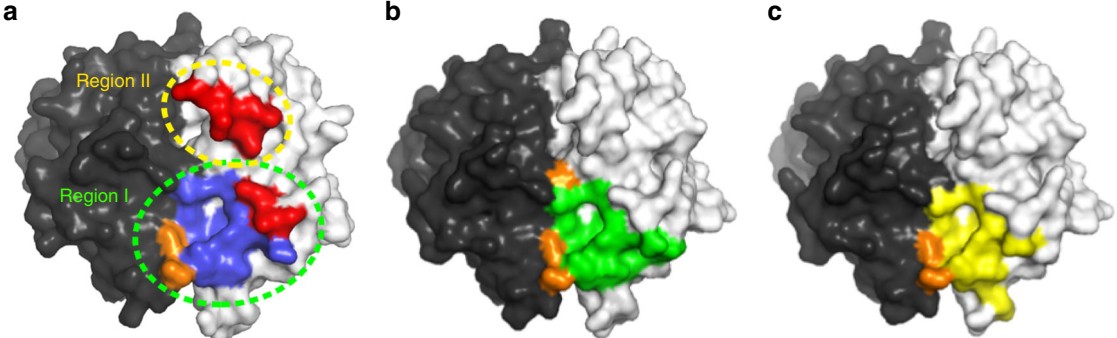

**Fig. 3** Comparison of belimumab epitope with receptor-binding sites. **a** The belimumab epitope on the BAFF surface. Two neighboring BAFF protomers are colored gray and black. The epitope regions for the heavy and light chains of belimumab are colored blue and red, respectively. The epitope region on an adjacent protomer, which is involved in the interaction with the belimumab heavy chain, is colored orange. Regions I and II are indicated as dotted ellipses. **b** The BR3-binding site is colored green and orange on the BAFF surface. **c** The BCMA-binding site is colored yellow and orange on the BAFF surface

and provides the key functional interaction with BAFF[7]. Interestingly, the conformation of the loop spanning from $_{heavy}$D101 to $_{heavy}$F105 within belimumab HCDR3 is very similar to those in BR3 (aa 26–30) and BCMA (aa 15–19), with shared sequence homology at $_{heavy}$D101, $_{heavy}$L103, and $_{heavy}$L104 of belimumab (Fig. 6). As a result, the belimumab HCDR3 loop binds BAFF in a manner similar to the interactions of BR3 and BCMA with BAFF. The DxL(V/L) motifs of BAFF receptors and belimumab interact with a highly focused pocket within BAFF region I, consisting of $_{BAFF}$Y206, $_{BAFF}$R231, $_{BAFF}$I233, and $_{BAFF}$R265 (Figs. 4a and 6a, b). The salt bridge and hydrogen bond mediated by $_{heavy}$D101 with $_{BAFF}$R265 and $_{BAFF}$Y206, as well as the hydrophobic interactions of $_{heavy}$L103 and $_{heavy}$L104 with $_{BAFF}$Y206 and $_{BAFF}$I233, precisely mimic the interactions between the DxL(V/L) motifs of BAFF receptors and BAFF, leading to antagonism of the ligand–receptor interaction.

The importance of the DxL(V/L) motif for BAFF binding has been investigated for therapeutic applications. Previous studies showed that a β-hairpin-structured peptide derived from the DxL (V/L) motif of BR3 blocked the BAFF–BR3 interaction and attenuated the disease process in a murine model of lupus[29,30]. Blisibimod also utilizes the key interaction associated with the DxL(V/L) motif for neutralizing BAFF[31]. Blisibimod is a peptibody, a Fc conjugated to a peptide consisting of four DxL(V/L) motifs, with high avidity to BAFF resulting from its tetravalency. Inversely, a previous study showed that an anti-BR3 antibody with sub-nM affinity mimics the structure of the BAFF region that interacts with the DxL(V/L) motif of BR3, thereby blocking the BAFF–BR3 interaction[32]. The crystal structure of the BAFF–belimumab complex reconfirmed the significance of the DxL(V/L) motif for the ligand–receptor interaction, suggesting that the highly focused pocket interacting with the DxL(V/L) motif might be a hot spot for discovery of small-molecule BAFF modulators that can complement anti-BAFF biologics.

## Discussion

BAFF exists in three forms: a membrane-bound form and two soluble forms that include a trimer and a 60-mer. Belimumab has been characterized as targeting soluble BAFF only, whereas other BAFF antagonists, including tabalumab, blisibimod, and atacicept, can block both membrane-bound and soluble BAFF[14]. Contrary to expectations, the biologics targeting both soluble and membrane-bound BAFF have failed to meet their primary endpoints in clinical trials of SLE. Although it is unclear whether the clinical consequences of BAFF antagonists correlate with the different mechanism of action against the various forms of BAFF, understanding the molecular basis of their distinct mechanisms

might provide important information for achieving clinical efficacy beyond that of belimumab, possibly through rational design of new biologics or combination therapies. In the crystal structures of TNF in complex with the Fab fragments of adalimumab, infliximab, and certolizumab, the orientations of these Fab fragments from the cell membrane were roughly similar to that of belimumab[33–35] (Supplementary Fig. 3). The binding angle and distance of belimumab Fab from the cell membrane was very close to those of adalimumab, and the structure of the TNF-certolizumab Fab complex showed an even more cramped space between the Fab fragment and the cell membrane. However, these anti-TNF antibodies are capable of blocking both soluble and membrane-bound TNF, whereas belimumab binds soluble BAFF only. Therefore, the inability of belimumab to bind membrane-bound BAFF appears to originate from an intrinsic property of BAFF, and not from its binding orientation or position. Accordingly, it is possible that the unique BAFF trimer–trimer interaction by the exceptionally long DE loop, which is involved in interactions with belimumab, might contribute to the inability of belimumab to bind membrane-bound BAFF, given that belimumab breaks the BAFF trimer–trimer interactions and disrupts the 60-mer, whereas tabalumab can bind both the 60-mer and the trimer, probably by maintaining the BAFF trimer–trimer interaction (Fig. 2). Although several studies have revealed that the major oligomeric states of soluble BAFF are the trimer and 60-mer, the oligomeric state of membrane-bound BAFF remains unknown. A previous study showed that cells expressing noncleavable membrane-bound BAFF were at least 50-fold more active than soluble BAFF trimers, although less active than the soluble 60-mer, implying that membrane-bound BAFF might exist as a higher-order oligomer other than a trimer[9,36]. On the basis of the crystal structure of the soluble BAFF 60-mer, it can be suggested that the BAFF trimer–trimer interaction in the 60-mer can also be formed between membrane-bound BAFF trimers using their flap regions. Therefore, the functional unit of membrane-bound BAFF could be a dimer, tetramer, pentamer, or higher oligomers of BAFF trimers through the trimer–trimer interactions, although the formation of the capsid-like 60-mer would not be feasible in the case of membrane-bound BAFF. Size-exclusion chromatography showed only trimers and 60-mers of soluble BAFF, but no other intermediate oligomers, suggesting that the trimer–trimer interaction is intrinsically weak and probably effective only in a cooperative way for the formation of the 60-mer[37]. This might be because freed soluble trimers can occupy nearly any position and orientation, yielding an increase in entropy. However, the putative trimer–trimer interaction between membrane-bound BAFF trimers would be stronger than that between soluble BAFF trimers,

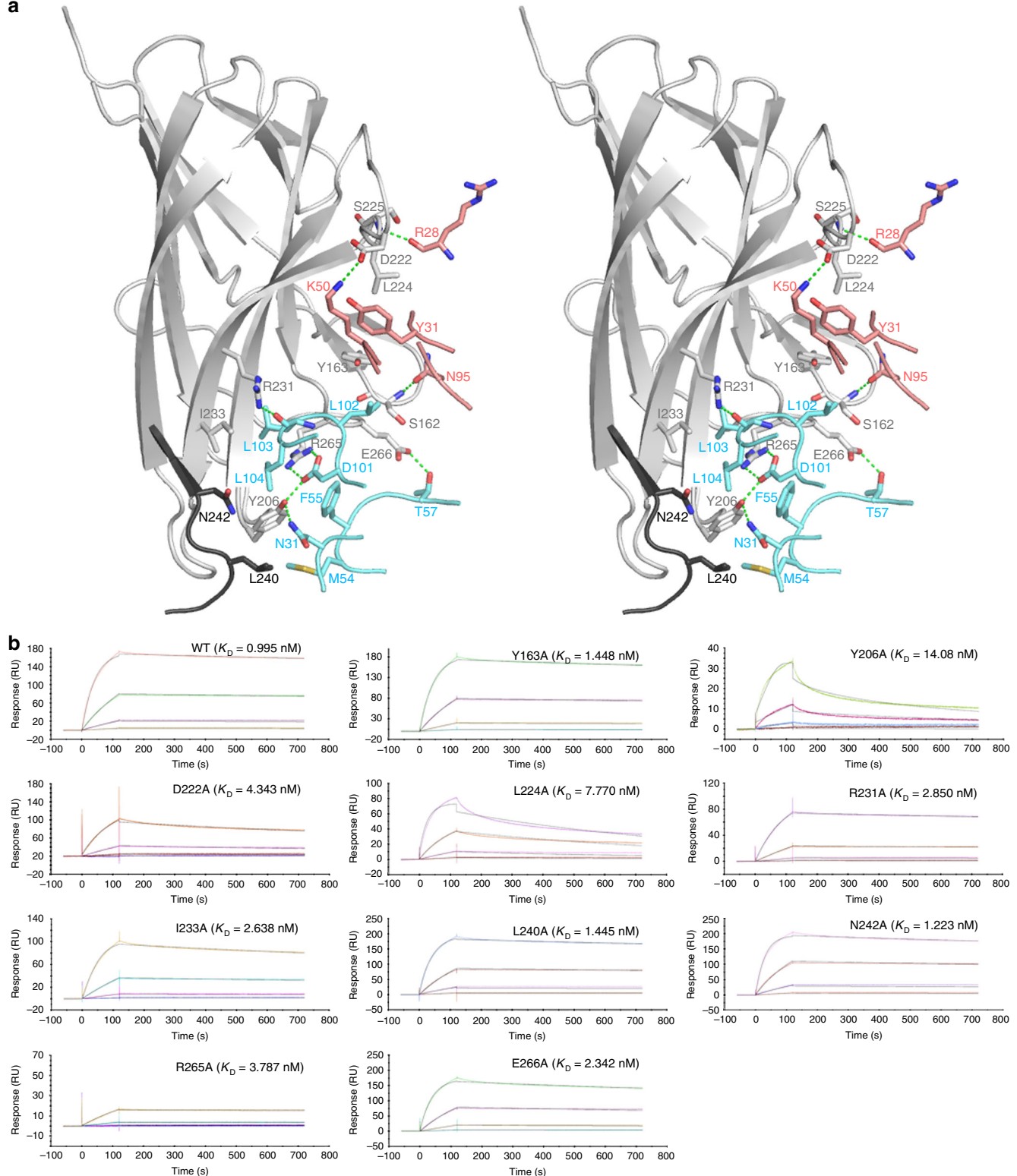

**Fig. 4** The BAFF–belimumab Fab interface. **a** Stereoview of the detailed BAFF–belimumab Fab interface. The carbon atoms from two neighboring BAFF protomers and the heavy and light chains of belimumab are colored gray, black, cyan, and pale red, respectively. Hydrogen bonds and salt bridges are indicated with dashed lines. **b** SPR sensorgrams for the binding kinetics of BAFF wild-type and mutants to the belimumab Fab. The concentrations of the wild-type and mutant BAFF for each experiment are 2, 10, 50, and 250 nM. For each experiment, one sensorgram out of two independent measurements with nearly identical results is shown

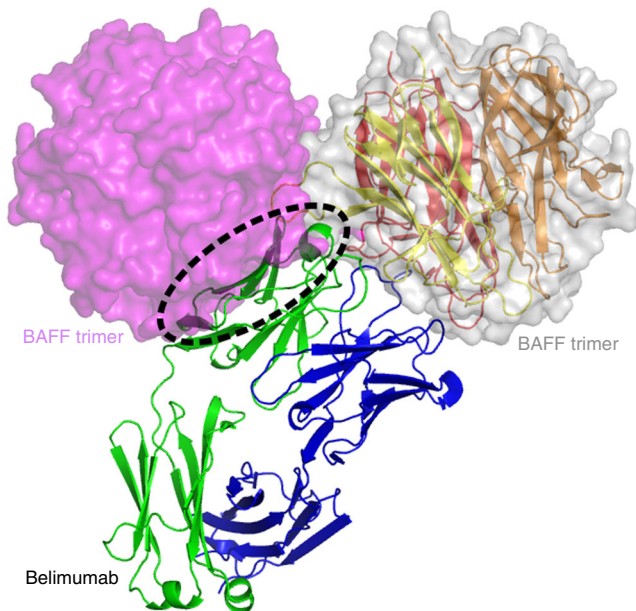

**Fig. 5** Blockade of the BAFF trimer–trimer interaction by belimumab. Partially transparent surface model of two neighboring BAFF trimers (purple and white) and a ribbon representation of one BAFF trimer (red, orange, and yellow for each protomer) in complex with belimumab Fab (heavy chain: blue; light chain: green). The dotted ellipse indicates a steric collision between the adjacent BAFF trimer and the bound belimumab

given that membrane-bound BAFF trimers are constrained to move on the cell membrane in close proximity to one another, and that their orientation on the membrane is also favorable for the BAFF trimer–trimer interaction via their flap regions. Therefore, the binding of belimumab to membrane-bound BAFF might be hindered by enhanced interactions between membrane-bound BAFF trimers, because belimumab binding would need to break the trimer–trimer interactions. The inhibitory capacities of belimumab and tabalumab against BAFF trimer and 60-mer were compared through NFκB luciferase report assays in CHO cells expressing either BR3 or TACI, and through B cell proliferation assays, showing unexpected differences in potency between belimumab and tabalumab against BAFF 60-mer[38]. Although both antibodies were nearly identical in their inhibitory capacity against BAFF trimer, tabalumab blocked the signaling effects of BAFF 60-mer at a strikingly lower concentration than belimumab. Considering that BAFF 60-mer is a stable assembly, belimumab could not block BAFF 60-mer until it reaches its critical concentration for disrupting the 60-mer, whereas occupation of only several epitopes exposed on the surface of the 60-mer by tabalumab may lead to the effective blocking of BAFF 60-mer without disrupting it. It remains unclear whether the successful approval for belimumab in contrast to tabalumab relates to the potential differences in the pharmacology between them, or whether it is based on the clinical study design including patient selection and end points.

A proliferation-inducing ligand (APRIL), the TNFSF member most closely related to BAFF, is able to bind TACI and BCMA,

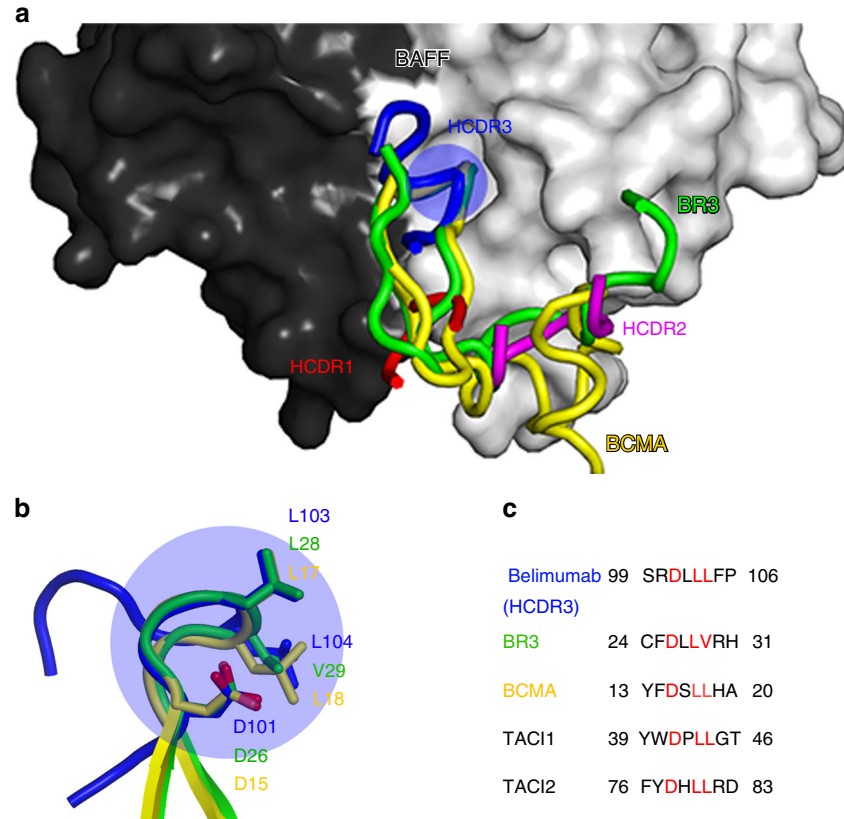

**Fig. 6** Similarity between belimumab HCDR3 and the DxL(V/L) motif. **a** Comparison of BAFF binding among BR3, BCMA, and the belimumab HCDR loops at the BAFF surface (black and gray for each protomer). **b** The conformations of the DxL(V/L) motifs within BR3 (green), BCMA (yellow), and the belimumab HCDR3 loop (blue). The key residues in the DxL(V/L) motifs are labeled. In **a** and **b**, the DxL(V/L) motifs are indicated by blue circles. **c** Amino acid sequence comparison between the belimumab HCDR3 loop and the DxL(V/L) motifs of BAFF receptors. Conserved residues are colored red

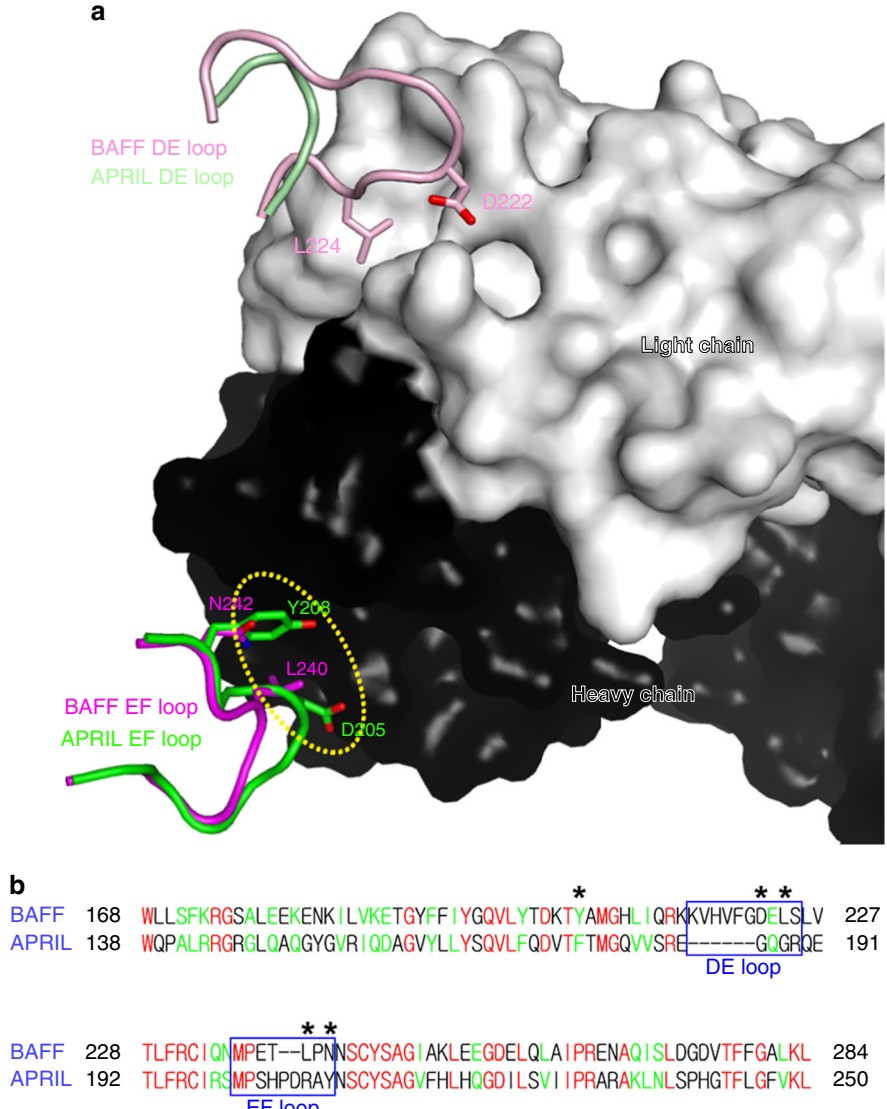

**Fig. 7** Structural basis of the selectivity of belimumab over APRIL. **a** The DE loops of BAFF and APRIL are colored pale purple and pale green, respectively. The residues $_{BAFF}$D222 and $_{BAFF}$L224, which are involved in the interaction with the belimumab light chain, are labeled. The EF loops in the adjacent BAFF and APRIL protomers are colored purple and green, respectively. The residues $_{BAFF}$L240 and $_{BAFF}$N242, which are involved in the interaction with the belimumab heavy chain, are labeled. The surfaces of the heavy and light chains of belimumab are colored black and white, respectively. Potential steric collision resulting from insertion within the EF loop of APRIL, especially by $_{APRIL}$D205 and $_{APRIL}$Y208, is represented by a dotted ellipse in the case of belimumab binding to APRIL. **b** Structure-based sequence alignment of BAFF and APRIL. The DE and EF loop regions are indicated with blue boxes. The BAFF residues involved in belimumab binding, but not conserved in APRIL, are indicated with asterisks. Identical and homologous residues are colored red and green, respectively

but not BR3[37]. As APRIL is also important for survival of plasmablasts and plasma cells, it has been suggested that blocking both BAFF and APRIL might have a greater neutralizing effect on autoantibodies than blocking BAFF alone. However, animal studies showed that APRIL was unnecessary for SLE pathogenesis, and that double knockout of BAFF and APRIL induced mild kidney immunopathology despite a greater decrease in the number of autoantibodies[39]. Furthermore, in a phase II/III clinical trial of atacicept, which is a TACI-Ig fusion protein that neutralizes both BAFF and APRIL, the increased toxicity outweighed any gains in efficacy[40]. The structure of the BAFF–belimumab complex provides the molecular basis for the selectivity of belimumab to block BAFF without neutralizing APRIL (Fig. 7). Despite the overall sequence homology between BAFF and APRIL, several key residues for belimumab binding are not conserved (Fig. 7b). The residue $_{APRIL}$F176, corresponding to

$_{BAFF}$Y206, would not allow formation of the important hydrogen bonds with $_{heavy}$N31 and $_{heavy}$D101 (Fig. 4a). The significantly reduced affinity of the BAFF mutant Y206A to belimumab also supports the suggestion that this amino acid difference would contribute to decreased affinity of belimumab for APRIL (Fig. 4b and Supplementary Table 1). The absence of the flap region in the DE loop of APRIL can also contribute to the BAFF selectivity of belimumab due to the lack of the key interactions by $_{BAFF}$D222 and $_{BAFF}$L224 with the belimumab light chain (Fig. 7). In addition, the insertion in the EF loop of APRIL would likely destroy the structural complementarity observed between the belimumab heavy chain and the residues of an adjacent protomer in the BAFF trimer ($_{BAFF}$L240 and $_{BAFF}$N242), resulting in steric hindrance that would preclude binding of belimumab to APRIL (Fig. 7). In addition to the homotrimers, human BAFF and APRIL can form heterotrimers with undefined stoichiometries,

and this formation seems to be increased in patients with auto-immune diseases[37]. A previous study with single-chain homo or heterotrimers of defined stoichiometry showed that BAFF and APRIL heterotrimers were differentially inhibited by belimu-mab[41]. Atacicept efficiently blocked both forms of the hetero-trimers, i.e., two BAFF and one APRIL ($B_2A_1$) and two APRIL and one BAFF ($A_2B_1$). In contrast, belimumab showed minimal inhibitory activity on $B_2A_1$, and no activity on $A_2B_1$. As belimumab binding would be negatively influenced by both of the two neighboring APRIL protomers in trimers, the $A_2B_1$ form would not be susceptible to belimumab binding due to the lack of the belimumab epitope, which comprises two adjacent BAFF protomers. By contrast, belimumab would bind the $B_2A_1$ form because one of three junctions of the heterotrimer contains the belimumab epitope. The very limited blocking activity of beli-mumab on $B_2A_1$ compared to BAFF homotrimer would be due to its lower affinity or avidity to $B_2A_1$ than the homotrimer. If a BAFF antagonist binds only a single BAFF protomer without being disturbed by a neighboring APRIL protomer, it could potentially block both forms of the heterotrimers, unlike belimumab.

Although therapeutic neutralization of BAFF with belimumab is efficacious in many SLE patients, a substantial percentage of SLE patients is unresponsive to this drug[42]. This modest effect of belimumab might be due to the great heterogeneity and complexity inherent to SLE. It is also possible that genetic variations or alterations in post-translational *N*-glycosylation status within BAFF might undermine the therapeutic capacity of belimumab in at least some of the non-responders. Human BAFF has two potential *N*-glycosylation sites, $_{BAFF}N124$ and $_{BAFF}N242$[3,8]. $_{BAFF}N124$ is not included in soluble BAFF, and $_{BAFF}N242$ is located on the opposite side of the belimumab epitope. Therefore, belimumab binding to BAFF should occur independently of BAFF glycosylation. However, genetic variations in BAFF can introduce new glycosylation sites, which might affect belimumab binding. A genomewide association study identified several insertion-deletion variants of BAFF, and it has been suggested that some of these BAFF variants could potentially contribute to SLE pathogenesis[43,44]. Although BAFF variants exhibiting reduced affinity to belimumab have not been reported, we believe that the structure of the BAFF–belimumab complex presented here would facilitate structure-based design of anti-BAFF agents that can avoid possible mutational escape of BAFF in the future.

Given that the epitope and mechanism of action are critical determinants for the development of therapeutic antibodies, our description of the binding modes revealed by the BAFF–belimumab complex structure in this study provides useful information for the discovery of improved BAFF antagonists and rational design of drug combinations with therapeutic synergy for the treatment of SLE and other autoimmune diseases in which BAFF has a central function in pathogenesis.

## Methods

**Expression and purification of BAFF.** Genes encoding the soluble form of human BAFF (aa 134–285) were subcloned into pET-28a (Novagen) (Supplementary Table 2). The protein was overexpressed with a N-terminal 6His-tag using plasmid-transformed *E. coli* BL21 (DE3) competent cells. The cells were first grown at 37 °C in LB medium supplemented with 50 μg mL$^{-1}$ Kanamycin. Protein expression was induced by adding 0.5 mM isopropyl β-D-1-thiogalactopyranoside (IPTG) when the cells reached an optical density at 600 nm of about 0.6, and the cells were grown for 16 h at 18 °C prior to harvesting by centrifugation (3000 × *g* for 0.5 h at 4 °C). The cell pellet was resuspended in a lysis buffer (20 mM HEPES, pH 7.0, 200 mM NaCl, 5 mM β-mercaptoethanol) and disrupted by sonication on ice. After the crude lysate was centrifuged (25,000 × *g* for 1 h at 4 °C), the supernatant containing soluble was applied to a HisTrap HP column (GE Healthcare Life Sciences) and washed with five column volumes of wash buffer (20 mM HEPES, pH 7.0, 200 mM NaCl, 5 mM β-mercaptoethanol, 50 mM imidazole). The protein was then eluted with elution buffer (20 mM HEPES, pH 7.0, 200 mM NaCl, 5 mM

β-mercaptoethanol, 400 mM imidazole). The eluted protein was treated with thrombin (Sigma-Aldrich) overnight at 277 K to remove the 6His-tag from the recombinant protein. After thrombin cleavage, the protein was concentrated and purified by gel-filtration chromatography using a HiLoad 16/60 Superdex 200 pg column (GE Healthcare Life Sciences) which had been pre-equilibrated with buffer containing 20 mM HEPES, pH 7.0, 200 mM NaCl. The elution profile of the protein showed two major peaks corresponding to the 60-mer and trimer. The eluted fractions for the trimer were collected, and the protein purity was evaluated by SDS–PAGE.

**Expression and purification of the Fab fragments.** The DNA sequences for the Fab fragments of belimumab and tabalumab were synthesized after codon-optimization for expression in *E. coli* (Bioneer, Inc). The sequences for the heavy chain and the light chain were cloned into pBAD-TOPO (Invitrogen), containing the STII signal sequence in each chain for periplasmic secretion and a C-terminal 6His-tag in the heavy chain (Supplementary Table 2)[45]. The plasmid pBAD-Fab was transformed into *E. coli* Top10F (Invitrogen). The cells were grown at 37 °C in LB medium supplemented with 50 μg mL$^{-1}$ ampicillin. At an OD$_{600}$ of 1.0, the protein expression was induced with 0.2% arabinose and cells were grown at 30 °C for 15 h. The cells were harvested by centrifugation, re-suspended in a lysis buffer (20 mM HEPES, pH 7.0, 200 mM NaCl), and lysed by sonication on ice. After removing cell debris by centrifugation (25,000 × *g* for 0.5 h at 4 °C), the supernatant containing soluble protein was applied to a HisTrap HP column (GE Healthcare Life Sciences) and washed with five column volumes of wash buffer (20 mM HEPES, pH 7.0, 200 mM NaCl, 50 mM imidazole). The protein was then eluted with elution buffer (20 mM HEPES, pH 7.0, 200 mM NaCl, 400 mM imidazole). The eluted protein was concentrated for gel-filtration chromatography using a HiLoad 16/60 Superdex 200 pg column (GE Healthcare Life Sciences). The column had previously been equilibrated with gel-filtration buffer (20 mM HEPES, pH 7.0, 200 mM NaCl). The elution profile of the protein showed a single major peak and the protein quality was evaluated by reducing and nonreducing SDS–PAGE.

**Analytical gel filtration.** The purified BAFF, belimumab Fab, and tabalumab Fab were analyzed using a Superdex 200 Increase 10/300 GL column (GE Healthcare Life Sciences) in a buffer containing 20 mM HEPES, pH 7.0, 200 mM NaCl. BAFF protein mixed with an excess amount of belimumab or tabalumab Fab was incubated for 1 h, and analyzed by size-exclusion chromatography.

**Crystallization and structure determination of belimumab Fab.** The purified belimumab Fab was concentrated to 9 mg mL$^{-1}$ in 20 mM HEPES, pH 7.0, and 200 mM NaCl. Crystals were grown using a hanging-drop vapor diffusion with a reservoir solution containing 0.1 M HEPES, pH 7.5, 0.2 M lithium sulfate mono-hydrate, and 25% (w/v) polyethylene glycol 3350 at 20 °C within a week. Crystals were cryoprotected by brief immersion in a well solution, supplemented with 20% ethylene glycol, and flash frozen in liquid nitrogen. X-ray diffraction data were collected at 100 K on beamline 7A of the Pohang Light Source (PLS) (Pohang, Korea). X-ray diffraction data were collected to a resolution of 1.90 Å, integrated, and scaled using HKL2000 (HKL Research). The structure was solved by molecular replacement using a Phaser[46] with a structure of the Fab fragments that has high sequence identities with belimumab Fab (PDB ID code 5BV7, chains H and L). Due to the intrinsic elbow flexibility of a Fab fragment, the Fv region and the other region including the CH1 and CL domains were separated when used as a search model. At this point, the electron density corresponding belimumab Fab was prominent. Iterative rounds of refinement were done using PHENIX[47] with manual inspection using COOT[48]. Statistics for data collection and refinement can be found in Table 1.

**Structure determination of the BAFF–belimumab Fab complex.** The purified BAFF and belimumab Fab were mixed in a 1:1.2 molar ratio and incubated for 1 h at 4 °C before being subjected to size exclusion chromatography using a HiLoad 16/60 Superdex 200 pg column equilibrated with 20 mM HEPES, pH 8.0, and 200 mM NaCl. Gel-filtration fractions containing the BAFF–belimumab Fab complex were concentrated to 8 mg mL$^{-1}$ in 20 mM HEPES, pH 8.0, and 200 mM NaCl. Crystals were grown using hanging-drop vapor diffusion with a reservoir solution containing 0.1 M Tris, pH 8.5, and 20% (w/v) polyethylene glycol 1000 at 20 °C within 15 days. Crystals were cryoprotected by brief immersion in the well solution, supplemented with 25% glycerol, and flash frozen in liquid nitrogen. X-ray diffraction data were collected at 100 K on beamline 5C of PLS. X-ray diffraction data were collected to a resolution of 2.05 Å, integrated, and scaled using HKL2000. The structure was solved by molecular replacement using Phaser with a structure of the free belimumab Fab fragment and one protomer of the BAFF 60-mer (PDB ID code 4V46). Iterative rounds of refinement were done using PHENIX with manual inspection using COOT. Statistics for data collection and refinement can be found in Table 1.

**Binding kinetics of the BAFF WT and mutants.** Site-directed mutants of BAFF, including $_{BAFF}Y163$, $_{BAFF}Y206$, $_{BAFF}D222$, $_{BAFF}L224$, $_{BAFF}R231$, $_{BAFF}I233$, $_{BAFF}L240$, $_{BAFF}N242$, $_{BAFF}R265$, and $_{BAFF}E266$, were created with the QuickChange Kit (Agilent Technologies, Catalog Code: 200518) and confirmed by DNA sequencing.

The mutant proteins were expressed and purified as described for wild-type BAFF. Approximately 1000 response units of the belimumab Fab fragment were immobilized on the surface of a CM-5 chip (GE Healthcare Life Sciences) via amine coupling reactions, as described in the manufacturer's instructions. Purified wild-type and the mutants of BAFF were dialyzed for 5 h against buffer A (20 mM sodium citrate, pH 5.6, 200 mM NaCl), and serially diluted to concentrations ranging from 2 to 250 nM using buffer A, and flowed through the chip. A BIAcore T100 instrument (GE Healthcare Life Sciences) was operated at 25 °C using buffer A as a running buffer. The bound BAFF was removed with 10 mM glycine (pH 2.0) at the end of each cycle while retaining the surface integrity for chip regeneration. Sensorgrams were locally fitted and the dissociation constants ($K_D$) were calculated with the analysis software, BIAevaluation (GE Healthcare Life Sciences). The SPR measurements were carried out in duplicate.

**Data availability**. The coordinates and structure factors for the crystal structures of the BAFF–belimumab Fab complex and the free belimumab Fab fragment have been deposited in the Protein Data Bank (www.rcsb.org) under accession codes 5Y9J and 5Y9K, respectively.

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

## Acknowledgements

We are grateful to the staffs of beamline 5C and 7A at Pohang Accelerator Laboratory for help with the X-ray diffraction experiments. This work was supported by grants from the National Research Foundation of Korea (NRF-2015R1D1A1A01057706 and NRF-2015M3A9B5030302).

## Author contributions

W.S. and Y.-S.H. designed research; W.S., H.T.L., H.L., S.H.L., J.Y.S., J.U.L., K.-Y.Y., J.R., J.Y.L., and Y.-S.H. performed experiments; W.S., H.T.L., H.L., S.H.L., S.E.R., and Y.-S.H. analyzed data; and Y.-S.H. wrote the paper.

## Additional information

**Competing interests:** The authors declare no competing interests.

