## [Peer Review File(PDF 188 kb) · Nature Communications]

Reviewers' comments:

Reviewer #1 (X-ray crystallography, antibody-antigen complexes)(Remarks to the Author):

The authors describe crystal structures of the monoclonal Fab Belimumab, and its complex with BAFF. In addition, they carried out structure-based mutagenesis studies of the BAFF and analyzed the interactions of the native and mutant proteins via SPR. The manuscript is clear, the crystallography appears to be sound, and the results are very interesting, with a well thought out analysis of the structures. I think anyone interested in Lupus therapies, antibody-antigen interactions, or tumor necrosis factor superfamily proteins will be interested in the results presented in this paper.

I have one minor complaint about the abstract, where Belimumab is described as having 'unprecedented therapeutic benefits'. Since the FDA says that it was only marginally effective, and you do comment on its 'modest effects' in your discussion section, I would recommend removal of the word 'unprecedented' in the abstract.

In Supp. Table 1 (the SPR results), were the measurements carried out in duplicate or triplicate? If so, please include the standard deviations. There are also a lot of significant figures, are these really valid?

Reviewer #2 (BAFF/APRIL, auto-antibody)(Remarks to the Author):

This manuscript presents an analysis of the BAFF/belimumab interactions strictly from a structural biological standpoint. The results point to belimumab inhibiting the activity of BAFF via two distinct pathways that involve two separate interactions between belimumab and BAFF.

First, belimumab blocks the epitope needed for binding of BAFF to its receptors. This will inhibit BAFF-mediated biological activity, as has previously been well documented.

Second, belimumab disassembles the BAFF 60-mers (which are presumed to have great biological potency) into trimers of BAFF/belimumab complexes (which are presumed to have less biological potency). Of note, tabalumab is incapable of disrupting BAFF 60-mers. This points to a biologically important difference between belimumab and tabalumab. It would greatly enhance the impact of this manuscript to provide experimental evidence that documents the biological relevance of this difference between belimumab and tabalumab.

An additional point (which the authors address in their Discussion) is that BAFF can form heterotrimers with APRIL. The authors postulate that belimumab may differentially bind to different BAFF/APRIL heterotrimers. If technically feasible, experimental testing of this possibility would also greatly enhance the impact of this manuscript.

Reviewer #1 (X-ray crystallography, antibody-antigen complexes)(Remarks to the Author):

The authors describe crystal structures of the monoclonal Fab Belimumab, and its complex with BAFF. In addition, they carried out structure-based mutagenesis studies of the BAFF and analyzed the interactions of the native and mutant proteins via SPR. The manuscript is clear, the crystallography appears to be sound, and the results are very interesting, with a well thought out analysis of the structures. I think anyone interested in Lupus therapies, antibody-antigen interactions, or tumor necrosis factor superfamily proteins will be interested in the results presented in this paper.

I have one minor complaint about the abstract, where Belimumab is described as having 'unprecedented therapeutic benefits'. Since the FDA says that it was only marginally effective, and you do comment on its 'modest effects' in your discussion section, I would recommend removal of the word 'unprecedented' in the abstract.

I agree.

In the revised abstract, 'unprecedented' was deleted to tone down.

In Supp. Table 1 (the SPR results), were the measurements carried out in duplicate or triplicate? If so, please include the standard deviations. There are also a lot of significant figures, are these really valid?

The SPR measurements were carried out in duplicate. In Figure 4, one sensorgram of each mutant out of two SPR experiments with nearly identical results is shown. The standard deviations are included in Supp. Table 1. In revision, the previous figure 3 was separated into two figures, figure 3 and 4, for clarity and impact.

Reviewer #2 (BAFF/APRIL, auto-antibody)(Remarks to the Author):

This manuscript presents an analysis of the BAFF/belimumab interactions strictly from a structural biological standpoint. The results point to belimumab inhibiting the activity of BAFF via two distinct pathways that involve two separate interactions between belimumab and BAFF.

First, belimumab blocks the epitope needed for binding of BAFF to its receptors. This will inhibit BAFF-mediated biological activity, as has previously been well documented.

Second, belimumab disassembles the BAFF 60-mers (which are presumed to have great biological potency) into trimers of BAFF/belimumab complexes (which are presumed to have less biological potency). Of note, tabalumab is incapable of disrupting BAFF 60-mers. This points to a biologically important difference between belimumab and tabalumab. It would greatly enhance the impact of this manuscript to provide experimental evidence that documents the biological relevance of this difference between belimumab and tabalumab.

Thank you for the suggestion.

In revision, we cited the paper (ref. #38), which highlighted the difference in the inhibitory potency between belimumab and tabalumab against BAFF 60-mer through in vitro cell-based assays. A paragraph related to this relevant document with our own opinion was presented in the revised discussion as follows.

“Recently, the inhibitory capacities of belimumab and tabalumab against

BAFF trimer and 60-mer were compared through NFκB luciferase report assays in CHO cells expressing either BR3 or TACI, and through B cell proliferation assays, showing unexpected differences in potency between belimumab and tabalumab against BAFF 60-mer³⁸. Although both antibodies were nearly identical in their inhibitory capacity against BAFF trimer, tabalumab blocked the signaling effects of BAFF 60-mer at a strikingly lower concentration than belimumab. Considering that BAFF 60-mer is a stable assembly, belimumab could not block BAFF 60-mer until it reaches its critical concentration for disrupting the 60-mer, whereas occupation of only several epitopes exposed on the surface of the 60-mer by tabalumab may lead to the effective blocking of BAFF 60-mer without disrupting it. It remains unclear whether the successful approval for belimumab in contrast to tabalumab relates to the potential differences in the pharmacology between them, or whether it is based on the clinical study design including patient selection and end points.”

An additional point (which the authors address in their Discussion) is that BAFF can form heterotrimers with APRIL. The authors postulate that belimumab may differentially bind to different BAFF/APRIL heterotrimers. If technically feasible, experimental testing of this possibility would also greatly enhance the impact of this manuscript.

We made great effort to obtain single-chain homo- or heterotrimers of BAFF and APRIL because of the intrinsic difficulty of producing and purifying the heterotrimers of defined stoichiometry. But, all our trials have failed. The expressed proteins were not stable, they aggregated completely. In revision, we cited the paper (ref. #41), which documents the differential inhibitory potency of belimumab against the different heterotrimers of BAFF and APRIL. This experimental result has much in common with our discussion and addresses your concern relevantly. A paragraph related to this document with our opinion for the

experiment was added in the revised discussion as follows.

“A previous study with single-chain homo- or heterotrimers of defined stoichiometry showed that BAFF and APRIL heterotrimers were differentially inhibited by belimumab⁴¹. Atacicept efficiently blocked both forms of the heterotrimers, i.e., two BAFF and one APRIL (B_2A_1) and two APRIL and one BAFF (A_2B_1). In contrast, belimumab showed minimal inhibitory activity on B_2A_1 , and no activity on A_2B_1 . As belimumab binding would be negatively influenced by both of the two neighboring APRIL protomers in trimers, the A_2B_1 form would not be susceptible to belimumab binding due to the lack of the belimumab epitope, which comprises two adjacent BAFF protomers. By contrast, belimumab would bind the B_2A_1 form because one of three junctions of the heterotrimer contains the belimumab epitope. The very limited blocking activity of belimumab on B_2A_1 compared to BAFF homotrimer would be due to its lower affinity or avidity to B_2A_1 than the homotrimer. If a BAFF antagonist binds only a single BAFF protomer without being disturbed by a neighboring APRIL protomer, it could potentially block both forms of the heterotrimers, unlike belimumab.”

REVIEWERS' COMMENTS:

Reviewer #2 (Remarks to the Author):

The authors have adequately addressed my previous concerns.

REVIEWERS' COMMENTS:

Reviewer #2 (Remarks to the Author):

The authors have adequately addressed my previous concerns.

I thank you for your comments.